# Photobiomodulation-Based Synergic Effects of Pt-Coated TiO_2_ Nanotubes and 850 nm Near-Infrared Irradiation on the Osseointegration Enhancement: In Vitro and In Vivo Evaluation

**DOI:** 10.3390/nano13081377

**Published:** 2023-04-15

**Authors:** Kyoung-Suk Moon, Ji-Myung Bae, Young-Bum Park, Eun-Joo Choi, Seung-Han Oh

**Affiliations:** 1Department of Dental Biomaterials and the Institute of Biomaterial and Implant, College of Dentistry, Wonkwang University, Iksan 54538, Republic of Korea; ksemoon@hanmail.net (K.-S.M.);; 2Department of Prosthodontology, College of Dentistry, Yonsei University, Seoul 03722, Republic of Korea; 3Department of Oral and Maxillofacial Surgery, College of Dentistry, Wonkwang University, Iksan 54538, Republic of Korea

**Keywords:** photobiomodulation, Pt, titania nanotubes, near-infrared, osseointegration

## Abstract

Photobiomodulation (PBM) therapy is known to have the potential to improve bone regeneration after implant surgery. However, the combinatory effect of the nanotextured implant and PBM therapy on osseointegration has not yet been proved. This study evaluated the photobiomodulation-based synergistic effects of Pt-coated titania nanotubes (Pt-TiO_2_ NT) and 850 nm near-infrared (NIR) light on osteogenic performance in vitro and in vivo. The FE-SEM and the diffuse UV-Vis-NIR spectrophotometer were used to perform the surface characterization. The live-dead, MTT, ALP, and AR assays were tested to perform in vitro tests. The removal torque testing, the 3D-micro CT, and the histological analysis were used to conduct in vivo tests. The live-dead and MTT assay resulted in Pt-TiO_2_ NTs being biocompatible. The ALP activity and AR assays demonstrated that the combination of Pt-TiO_2_ NT and NIR irradiation significantly enhanced osteogenic functionality (*p* < 0.05). The results of in vivo test, employing the removal torque testing, the 3D-micro CT, and histological analysis, showed overall improved outcomes; however, no significant difference was observed between the control and experimental groups (*p* > 0.05). Therefore, we confirmed the possibility of the combination of Pt-TiO_2_ NT and NIR light as a promising technology for implant surgery in dentistry.

## 1. Introduction

The successful osseointegration of dental implants is a critical aspect of implant surgery. Ideally, a healthy and solid bone should be formed around the implant in a short period. Various factors, such as implant surface treatment [1,2], bone morphogenic protein [3,4], and growth factors [5,6], have been investigated to promote osteogenesis around implants. However, conventional methods have several limitations; therefore, alternative methods that continuously improve osteogenesis are essential. Photoassisted implant osteogenesis enhancement methods primarily involve two approaches (1) activating the function of the implant surface via light irradiation, which is commonly known as photofunctionalization, and (2) accelerating the functions of cells or tissues around the implant using light, which is known as photobiomodulation (PBM) [7]. 

The photofunctionalization of titanium implants refers to the activation of the titanium implant surface using the ultraviolet (UV)-based photocatalytic effect of TiO_2_ present on the titanium surface. This results in excellent antibacterial activity and enhanced osseointegration of the implant [8,9,10]. PBM, which was previously known as low-level laser therapy, involves the application of visible light (>600 nm) in the near-infrared (NIR) range to improve wound healing and relieve acute and chronic pain [11,12,13]. Especially when considering the effect of PBM therapy in the clinical setting, several factors, such as the dose of light, the wavelength of light, and the safety from the light irradiation, play an essential role in the success of PBM therapy [14,15]. The recent development of optical technology has enabled PBM therapy using light-emitting diodes (LED) that have the specific wavelength characteristics of conventional lasers, in addition to the advantages of high outputs and ease of operation [16,17]. In dentistry, PBM therapy has been used for various clinical applications, such as treating hypersensitivity, craniofacial wound healing, and cancer therapy [18,19,20,21]. Moreover, in vivo studies have demonstrated the potential of PBM therapy in reducing healing time and improving bone regeneration after implant surgery [22,23]. The early application of PBM therapy in implant surgery has been shown to accelerate the proliferation and differentiation of osteoblasts and enhance the bonding between the titanium implant and the surrounding tissues [24,25,26]. Therefore, LED-based PBM is anticipated to significantly affect osteogenesis around dental implants.

In our previous research, we identified absorption patterns of light in the visible and NIR ranges based on the combination of noble metal nanoparticles and TiO_2_ nanotubes (NT) [27,28]. Specifically, the promotion of bone formation was confirmed under visible light at wavelengths of 470 and 600 nm (in the visible light region). However, thus far, no studies have examined the effect of combining noble metal nanoparticles and TiO_2_ NT with NIR light on osseointegration. 

Therefore, in this study, Pt-doped TiO_2_ NT with an excellent osteogenic function in the 470 and 600 nm visible light region was prepared. Additionally, the synergistic effect of noble metal doped TiO_2_ NT and NIR irradiation on the improvement of osseointegration both in vitro and in vivo was investigated. The null hypotheses of this study are that there are no significant differences in the results of (1) in vitro tests and (2) in vivo tests according to the presence or absence of Pt coating and 850 nm NIR light irradiation. 

## 2. Materials and Methods

### 2.1. Preparation and Characterization of Pt-Coated TiO_2_ NT (Pt-TiO_2_ NT)

TiO_2_ NT (diameter: 100 nm) was fabricated by the anodization (voltage: 20 V, duration: 30 min) of a pure Ti sheet (99.5%; Thickness 250 μm, Hyundai Titanium Co., Incheon, South Korea). A piece of Ti sheet (5 × 5 cm^2^) was anodized using hydrofluoric acid (0.5 *w*/*v*%, Merck & Co. Inc., Chicago, IL, USA). The anodized specimen was then dried (temperature: 60 °C, duration: 24 h) and heat-treated (Temperature: 400 °C, soaking time: 3 h) to crystalize the specimen. The heat-treated specimens were coated with Pt (coating time: 60 s) using an ion beam sputtering system (E-1030, Hitachi Co., Tokyo, Japan). The surface morphology and optical properties of the Pt-TiO_2_ NT were characterized by the field-emission scanning electron microscopy (FE-SEM; S-4800; Hitachi Co., Tokyo, Japan) and the diffuse reflectance UV–Vis–NIR spectrophotometer (SolidSpec-3700; Shimadzu Co., Kyoto, Japan). 

### 2.2. In Vitro Test

#### 2.2.1. Live-Dead Assay of Human Mesenchymal Stem Cells (hMSCs)

hMSCs (PT-2501, Lonza Co., Basel, Switzerland) were cultured in α-modified eagle’s minimum essential medium (α-MEM; Invitrogen, Carlsbad, CA, USA) with 10% fetal bovine serum (FBS; Invitrogen, Carlsbad, CA, USA) and 1% antibiotics (Invitrogen, Carlsbad, CA, USA) at 37 °C in a 5% CO_2_ incubator. To assist the osteogenic differentiation of hMSCs in a typical environment, 10 mM β-glycerolphosphate (Sigma, St. Louis, MO, USA), 50 µg/mL ascorbic acid (Sigma, St. Louis, MO, USA), and 10 nM 1α,25-dihydroxyvitamin D3 (Sigma, St. Louis, MO, USA) were added to the cell growth media. 

To perform the live/dead assay, hMSCs were dispensed at a concentration of 1 × 10^4^ cells/well in a 24-well plate containing the specimen (1 × 1 cm^2^). After 24 h of incubation, the hMSCs seeded specimen was under the irradiation of NIR light using a laboratory-fabricated LED (Wavelength: 850 nm, power density: 60 mW/cm^2^, the distance between the LED and the specimen: 4 cm, and the irradiation time: 15 min). After irradiation, the specimen was incubated for 24 and 48 h. To visualize live and dead cells, 500 μL of a phosphate-buffered solution (PBS; Gibco, Carlsbad, CA, USA) with 2 µM calcein AM (Invitrogen, Carlsbad, CA, USA) and 4 µM ethidium homodimer-1 (EthD-1, Invitrogen, Carlsbad, CA, USA) was added to each well after additional incubation for 24 and 48 h. After 30 min, live (green fluorescence color) and dead (red fluorescence color) cells were confirmed using an inverted fluorescence microscope (CKX41; Olympus Co., Tokyo, Japan).

#### 2.2.2. MTT Assay

Cell toxicity was assessed using an MTT assay kit (Sigma-Aldrich, St Louis, MO, USA). The same cell culture and NIR light irradiation conditions as those utilized for the live/dead assay were employed for the MTT assay. The testing and evaluation of the MTT assay were performed according to the protocol specified in ISO 10993-5, Annex C [29]. After an additional 24 and 48 h of incubation, 100 µg/mL of MTT solution was added to each well, and the samples were cultured at 37 °C in a 5% CO_2_ incubator. After 4 h, the formazan produced by the MTT solution was dissolved in DMSO (Sigma-Aldrich, St Louis, MO, USA), and the absorbance was measured at 570 nm using a microplate ELISA reader (Spectra MAX 250; Molecular Devices Co., Sunnyvale, CA, USA). If the cell viability value of the specimen is higher than 70% of the control (hMSCs cultured on the cell culture dish), the specimen is determined to be biocompatible according to the decision of ISO 10993-5. The MTT assay was performed on four samples from each group. 

#### 2.2.3. Alkaline Phosphatase (ALP) Activity Assay

The ALP activity assay was conducted utilizing the same cell culture and initial NIR light irradiation conditions as those of the live/dead assay. The hMSCs were continuously exposed to NIR light after changing the osteogenic media every three days. The ALP activity was measured after one and two weeks of incubation. The detailed procedure of ALP activity assay is the same as that of our previous study [27]. The absorbance was measured at 405 nm using a microplate ELISA reader, and the final ALP activity values were calculated by dividing the absorbance value by the average value of the total protein amount obtained from the experimental group. The ALP activity assay value of each group was determined by four samples of the group. 

#### 2.2.4. Alizarin Red (AR) Assay

The conditions for the cell culture and NIR light irradiation for hMSCs in the AR assay were identical to those used for the ALP activity assay. The AR was measured after two and three weeks of incubation. The detailed procedure of AR assay is the same as that of our previous study [27]. The absorbance was measured at 405 nm using a microplate ELISA reader. The AR assay value of each group was determined by four samples of the group. 

### 2.3. Animal Study

#### 2.3.1. Production of Experimental Animal Models and Implant Specimens

An in vivo study was conducted in accordance with the regulations of the Animal Experiment Ethics Committee of Wonkwang University (approval number: WKU21-20). Fifty male Sprague-Dawley rats (SD rats; body weight 250–280 g; 8 weeks old; Samtaco Co., Osan, Korea) were divided into two groups, namely, the uncoated TiO_2_ NT implant group as a control and the Pt-TiO_2_ NT group as the experimental group. Each group was further divided into two subgroups (four subgroups in total) based on the period of sacrifice (two and six weeks). A modified orthodontic screw (OSTH 1604, Osstem implant Co., Seoul, Korea) with a length of 4.0 mm and a diameter of 1.6 mm was utilized for the in vivo study. The length of the orthodontic screw was trimmed to 3 mm, which was the entire length of implantation, to fit the femur of the SD rats.

#### 2.3.2. Experimental Procedure of In Vivo Study

Figure 1 concisely depicts the experimental protocol employed in the in vivo study. Surgical procedures were performed under general anesthesia by administering 10 mg/kg Rompun (Bayer Korea Co., Seoul, Republic of Korea) and 30 mg/kg Zoletil (Virbac Lab, Carros, France). Ten minutes after the anesthesia injection, the surgical site was isolated and sterilized using a povidone–iodine solution. Subsequently, a 2% lidocaine injection was administered subcutaneously, and the surgical sites in the left and right femurs were drilled with a 2.0 mm diameter. Customized implants were then inserted into the osteotome sites until the head of the implants reached the cortical bone, and the surgical sites were sutured with 3.0 silk. Uncoated TiO_2_ NT and Pt-TiO_2_ NT implants were inserted into the right and left femurs, respectively. Three days after implantation surgery, the implanted animals were subjected to NIR irradiation in an LED chamber (20.0 × 12.0 × 24.0 cm^3^) for 15 min. NIR light irradiation was performed every three days. The experimental rats were euthanized in a CO_2_ gas chamber after two and six weeks of surgical intervention. Samples collected from the right femur were subjected to a removal torque test and histological analysis, whereas a sample collected from the left femur was subjected to radiological analysis. 

#### 2.3.3. Removal Torque Test

The sample obtained from the right femur, along with the implant, was placed in a removal torque testing system connected to a digital torque gauge (MTT03-50Z; Mark-10, New York, NY, USA). The upper notch of each implant was fixed to the Jacobian chuck of the digital torque gauge, and the chuck was rotated in the counterclockwise direction until the measured torque value reached the maximum value and went down. The maximum torque values were recorded during this process. The removal torque test value of each group was determined by five samples of the group. 

#### 2.3.4. Micro-Computed Tomography (CT) Assessment

To evaluate the implant placement position and peri-implant bone formation in the surrounding bone, the new bone volume ratio was calculated using micro-CT scanned images. The implant specimen in the mouse femur was scanned using a micro-CT system (Skyscan 1076, Bruker Co., Aartselaar, Belgium) at 100 kV and 100 µA at 700 ms intervals. At the time of measurement, the newly formed bone was defined as the region of interest (ROI), which was a cylindrical region surrounding the implant surface (0.4 mm in length and 1.1 mm in length). The new bone volume to total volume (BV/TV) was calculated for each group using data analysis software (CTAn, v1.9.1.0, Bruker Co., Aartselaar, Belgium). The new bone volume ratio was measured in five samples from each group.

#### 2.3.5. Histological Analysis

Following the removal torque test, the collected samples were preserved in 10% neutral buffered formalin at 4 °C for 14 d and subsequently placed in 1% EDTA (Sigma-Aldrich, St Louis, MO, USA) for 14 d to facilitate demineralization. Afterward, the samples were embedded in paraffin wax, and 4 μm thick sections were obtained using a rotary microtome (Shandon Finesse 325, Thermo Fisher Scientific, Waltham, MA, USA), parallel to the long axis of the femur. The obtained sections were stained with hematoxylin and eosin (H&E) to visualize the extracellular matrix and osteogenesis-related cells. The stained specimens were then analyzed using an optical microscope (CKX41, Olympus Co., Tokyo, Japan).

### 2.4. Data Analysis

All data in this study are expressed as mean ± standard deviation. Statistical analysis was performed on the data for all experiments using a one-way analysis of variance (IBM SPSS Statistics 24.0; IBM, Armonk, NY, USA), followed by the post hoc Games–Howell test. Differences were considered significant when *p*-values were less than 0.05.

## 3. Results

Figure 2 and Figure 3 depict the FE-SEM images and diffuse reflectance UV–Vis–NIR spectrophotometry results of TiO_2_ and Pt-TiO_2_ NT, respectively. The FE-SEM images resulted that the average diameters of TiO_2_ and Pt-TiO_2_ NT were 9.89 ± 1.68 and 12.76 ± 2.91 nm, respectively. Furthermore, Pt nanoparticles were located on the top surfaces of the NT. Based on the results of the diffuse reflectance UV–Vis–NIR analysis, the Pt-TiO_2_ NT group exhibited four electron absorption spectra, and one of them was detected within the 800–900 nm range, which was consistent with the wavelength of the NIR light utilized in this study. 

Figure 4 presents images of calcein-AM and EthD-1-stained hMSCs cultured on TiO_2_ and Pt-TiO_2_ NT, along with MTT assay results, with and without 850 nm NIR light irradiation. The images showed no evidence of damaged or dead cells (indicated by red fluorescence) in any of the groups after 24 and 48 h of cultivation. Moreover, the elongation ratio of the filopodia of hMSCs cultured on Pt-TiO_2_ NT specimen with 850 nm NIR light irradiation was higher than those of the other groups. The MTT assay results indicated no significant differences among all experimental groups (*p* > 0.05), and the cell viability values of all groups were above 70% compared to the control group (hMSCs cultured on cell culture dish), indicating good biocompatibility in all experimental groups.

Figure 5 reveals the ALP activity assay results of hMSCs under 850 nm NIR light irradiation. In the graph, (−), (+), and (850) indicate conditions without osteogenic media, with osteogenic media, and with combining osteogenic media and 850 nm NIR irradiation, respectively. After one week of cultivation, the Pt-TiO_2_ NT with 850 nm NIR light irradiation exhibited the highest ALP activity among all groups, and the activity was significantly higher than that observed for other groups and conditions (*p* < 0.05). After two weeks of cultivation, the ALP activity of the Pt-TiO_2_ NT with 850 nm NIR light irradiation was the highest among those of all groups. However, no significant difference was observed between TiO_2_ NT and Pt-TiO_2_ NT groups under 850 NIR light irradiation (*p* > 0.05).

Figure 6 displays the outcomes of the AR assay. In the graph, (−), (+), and (850) indicate conditions without osteogenic media, with osteogenic media, and with combining osteogenic media and 850 nm NIR irradiation, respectively. After three weeks of cultivation, the AR value of the Pt-TiO_2_ NT group with 850 nm NIR light irradiation was the highest among those of all groups and was significantly higher than those of other groups and irradiation conditions (*p* < 0.05). 

Figure 7 presents the results of the removal torque tests. In the graph, (850) means the condition with 850 nm NIR irradiation in animal experiments. Two weeks and six weeks after implantation, the torque value of the Pt-TiO_2_ NT group with 850 nm NIR light irradiation was the highest among those of all groups. However, there was no significant difference between the groups (*p* > 0.05). Furthermore, although the results of the removal torque test did not exhibit a statistically significant improvement, 850 NIR light irradiation enhanced the removal torque test results.

Figure 8 depicts the 3D micro-CT images of TiO_2_ NT and Pt-TiO_2_ NT with and without 850 nm NIR light irradiation. Following two and six weeks of implantation, widespread bone formation (yellow color in Figure 8) was observed surrounding the TiO_2_ NT or Pt-TiO_2_ NT surface-treated implant (white color in Figure 8), and the periosteum was elevated. Figure 9 presents the findings of the 3D micro-CT analysis, indicating a tendency for an increase in bone volume with 850 nm NIR irradiation. However, no statistically significant difference was observed depending on light irradiation (*p* > 0.05).

Figure 10 and Figure 11 show H&E-stained images of TiO_2_ NT and Pt-TiO_2_ NT at two- and six-week implantation, respectively. The thread-type implant–bone interfaces of most samples were disrupted after the removal torque test, so we observed only an intact bone-implant interface in any of the specimens. Therefore, we present a representative image of each group with the limited specimen conditions. Two weeks after implantation (Figure 10), we could not find a significant histological difference between TiO_2_ NT and Pt-TiO_2_ NT regardless of 850 nm NIR light irradiation. Instead, the immature bone matrix was observed at the implant-surrounding bone interface in both groups (black arrows in Figure 10). After six weeks of implantation (Figure 11), histological differences between the groups were not found regardless of 850 nm NIR light irradiation. However, mature new bone matrix, including osteocytes, appeared at the interface between the implant and existing bone (blue arrows in Figure 11).

## 4. Discussion

Based on all results of the in vitro and in vivo tests, the first null hypothesis established before the experiment was rejected, but the second null hypothesis was accepted. In vitro tests showed that the combination of Pt-TiO_2_ NT and 850 nm NIR light irradiation improved osteogenic performance. However, the results of in vivo tests indicated that this combination did not show any noticeable effect of enhancing osseointegration. 

Our previous study demonstrated that noble metal-coated TiO_2_ NT exhibits significant antibacterial activities through plasmonic photocatalytic effects [27,28]. Additionally, we demonstrated improved osteogenic capability when using implants with NT surface treatment and visible light irradiation. Specifically, the combination of Pt-coated TiO_2_ NT and 600 nm visible light irradiation used for the PBM treatment exhibited excellent osteogenic ability in vitro [27]. Based on these in vitro results, we planned to conduct in vivo experiments. However, we found that a skin penetration depth of 600 nm was inadequate for animal experiments. Therefore, we employed NIR radiation, which penetrates soft tissue more deeply than visible light and is an effective wavelength range for PBM therapy in conjunction with 600 nm visible light [30,31,32]. Therefore, in this study, we aimed to investigate the synergistic effects of NIR irradiation and Pt-TiO_2_ NT, which are known to promote wound healing, tissue repair, and osteogenesis. 

The inset of the FE-SEM image indicates that most of the Pt nanoparticles were deposited on the top surface of the TiO_2_ NT, exhibiting various shapes such as spherical, rod-like, and crescent-like morphology, owing to the features of the ion beam plasma coating system. Diffuse reflectance UV–Vis–NIR spectrophotometry analysis revealed four light absorbance peaks, with two significant peaks in the range of 550–650 and 800–900 nm, which are related to the photothermal scattering (short and long axes) of the deposited Pt nanoparticles. This result is due to the shape of the deposited Pt nanoparticles, as previously noted in other studies. We previously demonstrated that the morphology of Pt nanoparticles deposited on TiO_2_ NT influences the wavelength range of the light absorbance peaks through the photothermal scattering of the Pt nanoparticles [33,34,35,36]. 

Based on the results of the live-dead and MTT assays, we confirmed that 850 nm NIR light irradiation for 15 min did not cause cytotoxicity. Furthermore, the adhesion and differentiation of hMSCs cultured on Pt-TiO_2_ NT specimens under NIR light irradiation were further improved. In this study, we only tested the live-dead and MTT assays to determine the biocompatibility of the experimental specimen. Due to the diversity of the material, the limited cytotoxicity evaluation cannot evaluate the comprehensive cytotoxicity of the material. Therefore, cytotoxicity evaluation in various ways should be performed, and we will evaluate the cytotoxicity of the experimental specimen through various cytotoxicity evaluation methods [37]. In addition, in contrast to that of visible light, prolonged exposure to NIR light can lead to the evaporation of the cell culture media and the inevitable heating of the media. Therefore, we conducted a pilot test to measure the temperature changes in cell culture media, including experimental specimens, before and after 850 nm NIR irradiation for 15 min. The temperature of the cell culture media, the media with TiO_2_ NT, and the media with Pt-TiO_2_ NT increased by 2.3, 2.8, and 2.6 °C, respectively (detailed data are not shown). Therefore, no difference between the control and experimental groups was expected owing to the effect of NIR irradiation on hMSCs differentiation, despite the temperature increase.

Based on the results of the ALP activity and AR assays, which showed the highest values for the Pt-TiO_2_ NT group under 850 nm NIR light irradiation compared to those of the other groups, the enhanced osteogenic differentiation of hMSCs was closely related to the combination of Pt-TiO_2_ NT and NIR light. The morphology of hMSCs cultured on Pt-TiO_2_ NT with or without 850 nm NIR irradiation was estimated by assessing the elongation ratio of hMSCs from calcein-AM-stained images (Figure 4). After 48 h of incubation, the elongation ratio of hMSCs cultured on Pt-TiO_2_ NT and irradiated with 850 nm NIR was higher than those of the other groups and conditions. The elongation of mesenchymal stem cells plays a significant role in improving osteogenic differentiation, apart from the adhesion and proliferation of cells [38,39]. Therefore, the combination of Pt-TiO_2_ NT and 850 nm NIR irradiation resulted in the highest elongation of hMSCs in this study, which is closely related to early bone formation in vitro. 

The results of the removal torque test, micro-CT, and histological analysis indicated that the combination of 850 nm light irradiation and Pt-TiO_2_ NT did not improve significantly bone formation in vivo compared to in vitro. Significantly, several NIR light irradiation variables, such as the light intensity, irradiation time, and irradiation distance, are limited to produce the results of in vivo tests like those of in vitro tests. Additionally, because the SD rats used in the animal experiments were constantly moving, irradiating the femur (implanted place) with the same light intensity for a specific time was impossible. To minimize the variations in the aforementioned variables, a small chamber accommodating one SD rat was fabricated as shown in Figure 1, and NIR light irradiation was performed for 15 min in this chamber. Although the in vivo test was performed in an improved experimental environment, we could not obtain in vivo results similar to those obtained in vitro. Therefore, improvements in the research design limitations (equipment and light irradiation devices) for ensuring consistent light irradiation are necessary, and related experiments are currently in progress. In addition, in terms of clinical implication, we confirmed the possibility of combining Pt-TiO_2_ NT and NIR light as a promising technology for implant surgery. Additionally, we expect to try this combination for patients with exceptional cases such as immune deficiencies and evaluate the reparative mechanisms contributing to defining an osseointegration procedure [40].

## 5. Conclusions

Within the scope of this study, we confirmed that the combination of Pt-TiO_2_ NT and 850 nm NIR light irradiation facilitates excellent osteogenic performance in vitro and not in vivo because of the variations in the experimental environmental variables, such as the method of NIR light irradiation at the implant placement site and the management of experimental animals. Although additional improvements in animal test conditions are required, the combination of Pt-TiO_2_ NT and 850 nm NIR light irradiation exhibits the potential for enhancing osseointegration and is expected to serve as a foundation technology for developing novel implantable devices.

## Figures and Tables

**Figure 1 nanomaterials-13-01377-f001:**
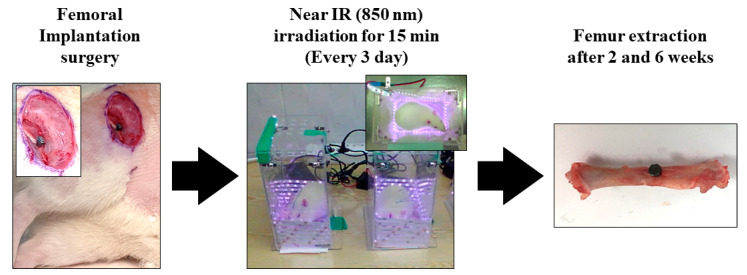
The experimental procedure of in vivo test.

**Figure 2 nanomaterials-13-01377-f002:**
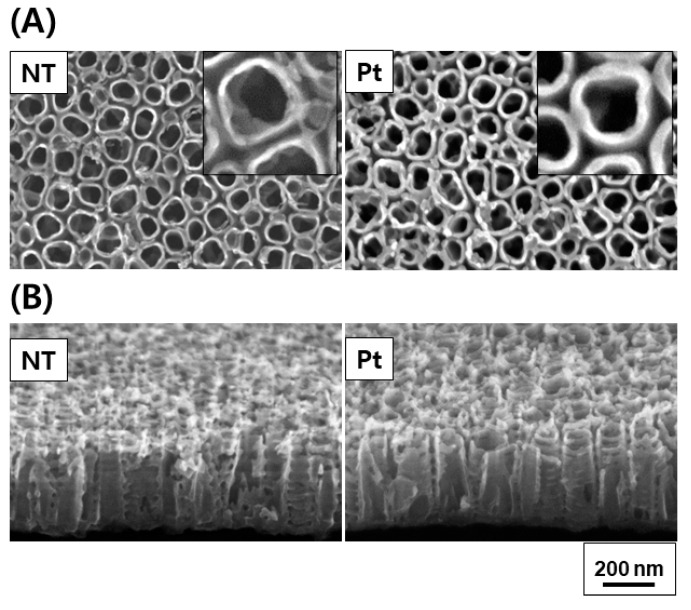
FE-SEM plain (**A**) and oblique views (**B**) of TiO_2_ NT and Pt-TiO_2_ NT (inset: magnified plain view).

**Figure 3 nanomaterials-13-01377-f003:**
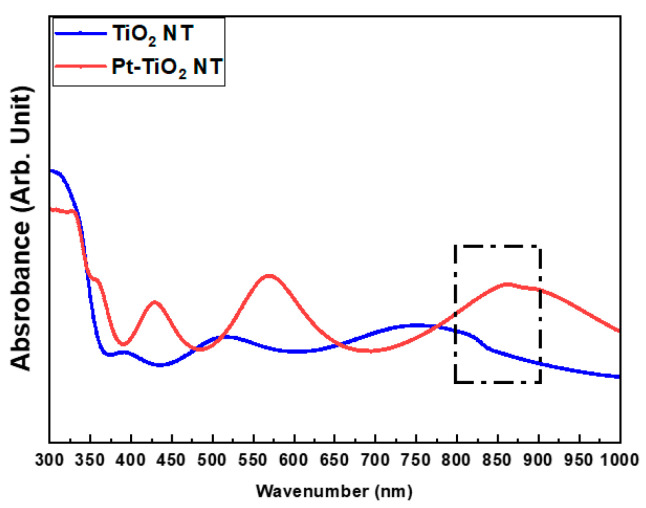
Diffuse reflectance UV–Vis–NIR spectrophotometry results of TiO_2_ NT and Pt-TiO_2_ NT (sputtering time: 60 s).

**Figure 4 nanomaterials-13-01377-f004:**
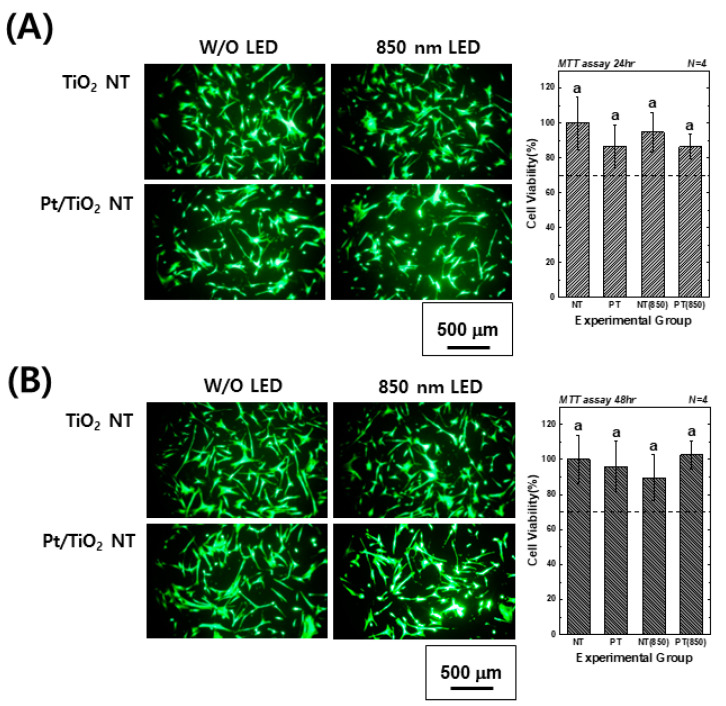
Calcein-AM- and EthD-1-stained images and MTT assay results of hMSCs cultured on TiO_2_ NT and Pt-TiO_2_ NT after (**A**) 24 and (**B**) 48 h of incubation (In each graph, the experimental groups with the same lowercase letters indicate statistical significance by one-way ANOVA at α = 0.05).

**Figure 5 nanomaterials-13-01377-f005:**
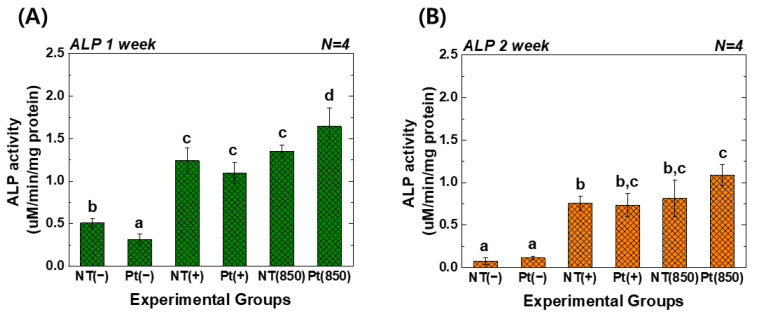
ALP activity values of TiO_2_ NT and Pt-TiO_2_ NT after (**A**) one week and (**B**) two weeks of incubation (In each graph, the experimental groups with different letters indicate statistical significance by one-way ANOVA at α = 0.05).

**Figure 6 nanomaterials-13-01377-f006:**
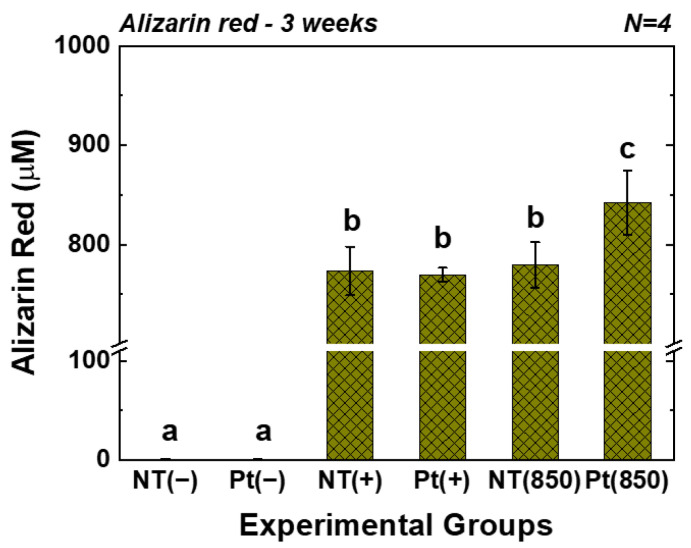
AR values of TiO_2_ NT and Pt-TiO_2_ NT after three weeks of incubation (In each graph, the experimental groups with different letters indicate statistical significance by one-way ANOVA at α = 0.05).

**Figure 7 nanomaterials-13-01377-f007:**
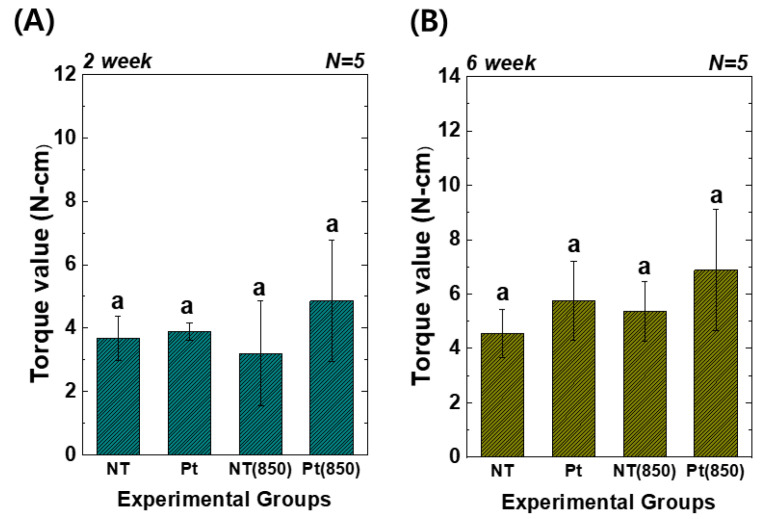
Torque test results of TiO_2_ NT- and Pt-TiO_2_ NT-treated implants after (**A**) two and (**B**) six weeks of implantation (In each graph, the experimental groups with different letters indicate statistical significance by one-way ANOVA at α = 0.05).

**Figure 8 nanomaterials-13-01377-f008:**
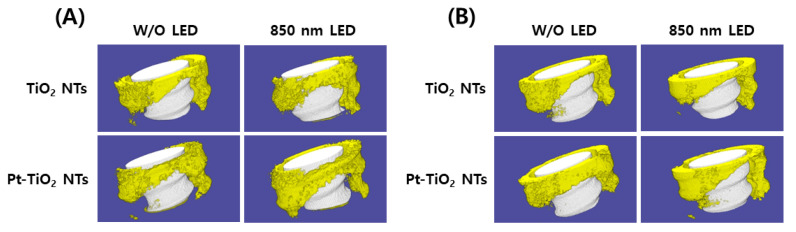
Three-dimensional micro-CT images and analysis of TiO_2_ NT- and Pt-TiO_2_ NT-treated implants after (**A**) two and (**B**) six weeks of implantation (white color: TiO_2_ NT or Pt-TiO_2_ NT surface-treated implant, yellow color: bone).

**Figure 9 nanomaterials-13-01377-f009:**
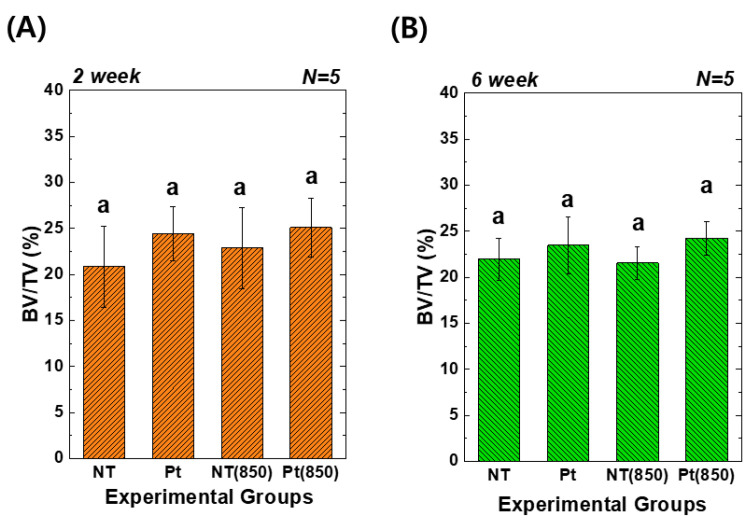
Three-dimensional micro-CT analysis of new bone volume (%) around the TiO_2_ NT- and Pt-TiO_2_ NT-treated implants after (**A**) two and (**B**) six weeks of implantation (In each graph, the experimental groups with the same lowercase letters indicate statistical significance by one-way ANOVA at α = 0.05).

**Figure 10 nanomaterials-13-01377-f010:**
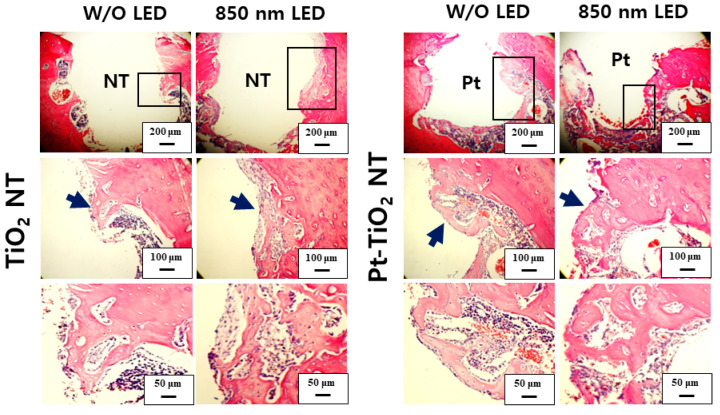
H&E-stained images of the tissues around the TiO_2_ NT- and Pt-TiO_2_ NT-treated implants after two weeks.

**Figure 11 nanomaterials-13-01377-f011:**
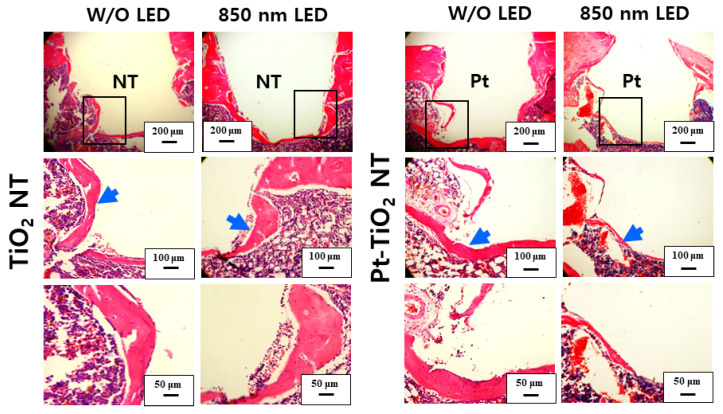
H&E-stained images of the tissues around the TiO_2_ NT- and Pt-TiO_2_ NT-treated implants after six weeks.

## Data Availability

The data presented in this study are available on request from the corresponding author.

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
