# Peer review of "Photobiomodulation-Based Synergic Effects of Pt-Coated TiO2 Nanotubes and 850 nm Near-Infrared Irradiation on the Osseointegration Enhancement: In Vitro and In Vivo Evaluation"

_nanomaterials, 2023, doi:10.3390/nano13081377_

Round 1

Reviewer 1 Report

Authors have done very interesting work. Please list Micro-CT scan in the abstract.

Suppose the readers are material scientists who have modest knowledge to read results of medical device. Please point the colour yellow or white for tissue or  TiO2 NT Pt-TiO2 NT in figure legend of the image of figure 8.

The figure 10 

Please label T or Pt in each image to indicate which side the material were contacted with bone tissue. Please use scale bar instead of magnification. 

Please descript what can we see. I may be obvious for an orthopedic, but most material scientists would not know  what is immature bone matrix. Please descript the different cells and what are different between the images of figure 10

Author Response

Reviewer 1 comments

Authors have done very interesting work. Please list Micro-CT scan in the abstract.

Suppose the readers are material scientists who have modest knowledge to read results of medical device. Please point the colour yellow or white for tissue or TiO2 NT Pt-TiO2 NT in figure legend of the image of figure 8.

è Thank you for the good suggestion. We added the description of white and yellow colors in the manuscript and the caption of Figure 8. The yellow highlighted phrase is the corrected one. (Line 271-272, Line 278-279)

Figure 8 depicts the 3D micro-CT images of TiO2 NT and Pt-TiO2 NT with and without 850 nm NIR light irradiation. Following two and six weeks of implantation, widespread bone formation (yellow color in Figure 8) was observed surrounding the TiO2 NT or Pt-TiO2 NT surface-treated implant (white color in Figure 8), and the periosteum was elevated. Figure 9 presents the findings of the 3D micro-CT analysis, indicating a tendency for an increase in the bone volume with 850 nm NIR irradiation. However, no statistically significant difference was observed depending on light irradiation (P > 0.05).

Figure 8. Three-dimensional micro-CT images and analysis of TiO2 NT- and Pt-TiO2 NT-treated implants after (A) two and (B) six weeks of implantation (White color: TiO2 NT or Pt-TiO2 NT surface-treated implant, yellow color: bone)

The figure 10 

Please label T or Pt in each image to indicate which side the material were contacted with bone tissue. Please use scale bar instead of magnification. 

Please descript what can we see. I may be obvious for an orthopedic, but most material scientists would not know what is immature bone matrix. Please descript the different cells and what are different between the images of figure 10

è We inserted ‘NT” and ‘Pt’ in each image and changed from magnification to scale bar.

è We separated Histological images into two Figures (10 and 11) according to the implantation periods. In addition, we added black arrows indicating an immature bone matrix (Figure 10) and blue arrows indicating a mature new bone area (Figure 11). Also, we added highly magnified images to Figure 10 and Figure 11 (the image with 50 mm of scale bar)

è Overall, we revised the explanation of the results of the histological analysis. (Line 287-294)  

Two weeks after implantation (Figure 10), we could not find a significant histological difference between TiO2 NT and Pt-TiO2 NT regardless of 850 nm NIR light irradiation. Instead, the immature bone matrix was observed at the implant-surrounding bone interface in both groups (Black arrows in Figure 10). After six weeks of implantation (Figure 11), histological differences between the groups were not found regardless of 850 nm NIR light irradiation. However, mature new bone matrix, including osteocytes, appeared at the interface between the implant and existing bone (Blue arrows in Figure 11).

Reviewer 2 Report

Interesting study well performed in its experimental part but still lacking in the definition of the general aspects. Here are the major criticisms: - in the initial part of the abstract section, an initial sentence on the problem that led to the study should be added -line 14 add by listing the tests performed - insert a final sentence on the clinical consequences of the study -check that all keywords are pubmed mesh terms - some considerations on the effects of LLLT in the clinical setting should be added - insert at the end of the introduction section the null hypotheses of the study which must be refuted at the end of the results obtained - define why the two evaluation times of cell viability were selected -in figure 4 insert the statistical significance of the results with an asterisk - a section on the limitations of the study is missing - some considerations on the use of cytotoxicity assays in general with advantages and disadvantages of the various methods should be added. In this regard, I ask you to insert in the reference section the following scientific work which could be of help to the reader: Pagano S, Lombardo G, Caponi S, et al. Bio-mechanical characterization of a CAD/CAM PMMA resin for digital removable prostheses. Dent Mater. 2021;37(3):e118-e130. doi:10.1016/j.dental.2020.11.003 - another aspect to consider is some considerations on the reparative mechanisms that contribute to defining an osseointegration procedure, particularly in patients with immune deficiencies. In this regard, I ask you to include the following scientific work in the reference section: Nesti M, Carli E, Giaquinto C, Rampon O, Nastasio S, Giuca MR. Correlation between viral load, plasma levels of CD4 - CD8 T lymphocytes and AIDS-related oral diseases: a multicenter study on 30 HIV+ children in the HAART era. J Biol Regul Homeost Agents. 2012;26(3):527-5

Author Response

Reviewer 2 comments

Interesting study well performed in its experimental part but still lacking in the definition of the general aspects. Here are the major criticisms:

- in the initial part of the abstract section, an initial sentence on the problem that led to the study should be added

è We added basic information on PBM therapy in dentistry and the motivation of this study in the Abstract.

-line 14 add by listing the tests performed

è We added the name of all tests in the Abstract.

- insert a final sentence on the clinical consequences of the study

à We added a phrase about the clinical applicability of the results of this study to the final sentence of the Abstract.

Abstract: Photobiomodulation (PBM) therapy is known to have the potential to improve bone regeneration after implant surgery. However, the combinatory effect of the nanotextured implant and PBM therapy on osseointegration has not yet been proved. This study evaluated the photobiomodulation-based synergistic effects of Pt-coated titania nanotubes (Pt-TiO2 NT) and 850 nm near-infrared (NIR) light on osteogenic performance in vitro and in vivo. The FE-SEM and the diffuse UV-Vis-NIR spectrophotometer were used to perform the surface characterization. The live-dead, MTT, ALP, and AR assays were tested to perform in vitro tests. The removal torque testing, the 3D-micro CT, and the histological analysis were used to conduct in vivo tests. The live-dead and MTT assay resulted that Pt-TiO2 NTs were biocompatible. The ALP activity and AR assays demonstrated that the combination of Pt-TiO2 NT and NIR irradiation significantly enhanced osteogenic functionality (P < 0.05). However, the results of in vivo test, employing the removal torque testing, the 3D-micro CT, and histological analysis, showed overall improved outcomes; however, no significant difference was observed between the control and experimental groups (P>0.05). Therefore, we confirmed the possibility of the combination of Pt-TiO2 NT and NIR light as a promising technology for implant surgery in dentistry. 

-check that all keywords are pubmed mesh terms

è We confirmed ‘photobiomodulation,’ ‘Pt,’ ‘near-infrared,’ and ‘osseointegration’ in PubMed MsSH. However, we could not find ‘titania nanotubes’ in PubMed MeSH, but we found this in PubMed and PubMed Central. Therefore, even though ‘titania nanotubes’ are not in the list of PubMed MeSH, we want to keep the term in Keyword because ‘titania nanotubes’ is the main Keyword in this study.

Keywords: photobiomodulation; Pt, titania nanotubes; near-infrared; osseointegration

- some considerations on the effects of LLLT in the clinical setting should be added

è We added some consideration on the effects of PBM therapy in the clinical setting in the Introduction. (Line 46-49)  

Especially when considering the effect of PBM therapy in the clinical setting, several factors, such as the dose of light, the wavelength of light, and the safety from the light irradiation, play an essential role in the success of PBM therapy

- insert at the end of the introduction section the null hypotheses of the study which must be refuted at the end of the results obtained

è We added the null hypotheses of this study to the last sentence of the Introduction. (Line 70-72)

. The null hypotheses of this study are that there are no significant differences in the results of (1) in vitro tests and (2) in vivo tests according to the presence or absence of Pt coating and 850 nm NIR light irradiation.

è In addition, we added the determination of the null hypotheses of this study to the first sentence of the Discussion. (Line 302-306)

Based on all results of the in vitro and in vivo tests, the first null hypothesis established before the experiment was rejected, but the second null hypothesis was accepted. In vitro tests showed that the combination of Pt-TiO2 NT and 850 nm NIR light irradiation improved osteogenic performance. However, the results of in vivo tests indicated that this combination did not show any noticeable effect of enhancing osseointegration.

- define why the two evaluation times of cell viability were selected

è The testing and evaluation of the MTT assay were performed according to the protocol specified in ISO 10993-5, Annex C. Therefore, we selected two evaluation times (24 and 48 hr). Also, we determined the biocompatibility of the specimen based on the decision described in ISO 10993-5. We added a detailed explanation relating to the MTT assay at 2.2.2. MTT assay. The yellow highlighted sentence is the corrected one. (Line 111-112, Line 117-119)

2.2.2. MTT assay

Cell toxicity was assessed using an MTT assay kit (Sigma-Aldrich, St Louis, MO, USA). The same cell culture and NIR light irradiation conditions as those utilized for the live/dead assay were employed for the MTT assay. The testing and evaluation of the MTT assay were performed according to the protocol specified in ISO 10993-5, Annex C [29]. After additional 24 and 48 h of incubation, 100 mg/mL of MTT solution was added to each well, and the samples were cultured at 37°C in a 5% CO2 incubator. After 4 h, the formazan produced by the MTT solution was dissolved in DMSO (Sigma-Aldrich, St Louis, MO, USA), and the absorbance was measured at 570 nm using an ELISA reader (Spectra MAX 250; Molecular Devices Co., Sunnyvale, CA, USA). If the cell viability value of the specimen is higher than 70% of the control (hMSCs cultured on the cell culture dish), the specimen is determined to be biocompatible according to the decision of ISO 10993-5. The MTT assay was performed on four samples from each group.

- in figure 4 insert the statistical significance of the results with an asterisk

è From the results of the MTT assay after 24 and 48 hr of hMSCs incubation, there is no significant difference between all groups (P>0.05). We added the alphabet ‘a’ to the graph to describe the results of the statistical analysis. Also, we mentioned the results of the statistical analysis of the MTT assay. The yellow highlighted sentence is the corrected one. (Line 228-229)

Figure 4 presents images of calcein-AM- and EthD-1-stained hMSCs cultured on TiO2 and Pt-TiO2 NT, along with MTT assay results, with and without 850 nm NIR light irradiation. The images showed no evidence of damaged or dead cells (indicated by red fluorescence) in any of the experimental groups after 24 and 48 h of cultivation. Moreover, the elongation ratio of the filopodia of hMSCs cultured on Pt-TiO2 NT under 850 nm NIR light irradiation was higher than those of the other groups. The MTT assay results indicated no significant differences among all experimental groups (P > 0.05), and the cell viability values of all experimental groups were above 70% compared to control group (hMSCs cultured on cell culture dish), indicating good biocompatibility in all experimental groups.

- a section on the limitations of the study is missing

è We added the limitation of the study to the last paragraph of the Discussion. The yellow highlighted sentence is the corrected one. (Line 360-362, Line 369-372)

The results of the removal torque test, micro-CT, and histological analysis indicated that the combination of 850 nm light irradiation and Pt-TiO2 NT did not improve significantly bone formation in vivo compared to in vitro. Significantly, several NIR light irradiation variables, such as the light intensity, irradiation time, and irradiation distance, are limited to produce the results of in vivo tests like those of in vitro tests. Additionally, because the SD rats used in the animal experiments were constantly moving, irradiating the femur (implanted place) with the same light intensity for a specific time was impossible. To minimize the variations in the aforementioned variables, a small chamber accommodating one SD rat was fabricated as shown Figure 1, and NIR light irradiation was performed for 15 min in this chamber. Although the in vivo test was performed in an improved experimental environment, we could not obtain in vivo results similar to those obtained in vitro. Therefore, improvements in the research design limitations (equipment and light irradiation devices) for ensuring consistent light irradiation are necessary, and related experiments are currently in progress.

- some considerations on the use of cytotoxicity assays in general with advantages and disadvantages of the various methods should be added.

In this regard, I ask you to insert in the reference section the following scientific work which could be of help to the reader: Pagano S, Lombardo G, Caponi S, et al. Bio-mechanical characterization of a CAD/CAM PMMA resin for digital removable prostheses. Dent Mater. 2021;37(3):e118-e130. doi:10.1016/j.dental.2020.11.003

è We explained some considerations on the use of cytotoxicity assays in general with the advantages and disadvantages of the various methods. Also, we inserted the reference (Line 333-337)

Due to the diversity of the material, the limited cytotoxicity evaluation cannot evaluate the comprehensive cytotoxicity of the material. Therefore, cytotoxicity evaluation in various ways should be performed, and we will evaluate the cytotoxicity of the experimental specimen through various cytotoxicity evaluation methods [37].

- another aspect to consider is some considerations on the reparative mechanisms that contribute to defining an osseointegration procedure, particularly in patients with immune deficiencies.

In this regard, I ask you to include the following scientific work in the reference section: Nesti M, Carli E, Giaquinto C, Rampon O, Nastasio S, Giuca MR. Correlation between viral load, plasma levels of CD4 - CD8 T lymphocytes and AIDS-related oral diseases: a multicenter study on 30 HIV+ children in the HAART era. J Biol Regul Homeost Agents. 2012;26(3):527-5

è We added the sentence that referred to another clinical implication of the combination of Pt-TiO2 NT and 850 nm NIR light irradiation as the reviewer’s suggestion. In addition, we added the reference (Line 373-375)

Also, we expect to try this combination for patients with exceptional cases like immune deficiencies and evaluate the reparative mechanisms contributing to defining an osseointegration procedure [40]

Round 2

Reviewer 1 Report

The manuscript has been improved.